# Is High-Risk Sexual Behavior a Risk Factor for Oropharyngeal Cancer?

**DOI:** 10.3390/cancers15133356

**Published:** 2023-06-26

**Authors:** Gunnar Wichmann, Jasmin Rudolph, Sylvia Henger, Christoph Engel, Kerstin Wirkner, John Ross Wenning, Samira Zeynalova, Susanne Wiegand, Markus Loeffler, Theresa Wald, Andreas Dietz

**Affiliations:** 1Department of Otorhinolaryngology, Head and Neck Surgery, University Hospital Leipzig, Liebigstr. 10-14, 04103 Leipzig, Germany; jasmin.rudolph@medizin.uni-leipzig.de (J.R.); johnross.wenning@medizin.uni-leipzig.de (J.R.W.); theresa.wald@medizin.uni-leipzig.de (T.W.); andreas.dietz@medizin.uni-leipzig.de (A.D.); 2LIFE Leipzig Research Center for Civilization Diseases, Leipzig University, Philipp-Rosenthal-Str. 27, 04103 Leipzig, Germany; sylvia.henger@imise.uni-leipzig.de (S.H.); christoph.engel@imise.uni-leipzig.de (C.E.); kerstin.wirkner@medizin.uni-leipzig.de (K.W.); samira.zeynalova@imise.uni-leipzig.de (S.Z.); markus.loeffler@imise.uni-leipzig.de (M.L.); 3Institute for Medical Informatics, Statistics and Epidemiology (IMISE), Leipzig University, Härtelstr. 16-18, 04107 Leipzig, Germany

**Keywords:** head and neck cancer (HNC), head and neck squamous cell carcinoma (HNSCC), oropharyngeal squamous cell carcinoma (OPSCC), human papillomavirus (HPV), sexual behavior (SB)

## Abstract

**Simple Summary:**

Several lines of evidence established a link between high-risk (HR) sexual behavior (SB), the persistence of human papillomavirus (HPV) DNA in saliva, and the presence of oncogenic HR-HPV subtypes in oropharyngeal squamous cell carcinoma (OPSCC). Especially one influential case-control study by D’Souza et al. was responsible for the definitive acceptance of “high risk sexual behavior” as being causatively involved in the etiology of HPV-driven OPSCC. Utilizing case-control studies can be problematic in respect to achieving reliable statistical inference. For generalizability and drawing conclusions for the general population, the selection of cases and controls studied is critical. Substantial bias can be introduced. Therefore, the aim of our study was to replicate these former findings in a nested case-control study of OPSCC patients and propensity score (PS)-matched unaffected controls from a large population-based German cohort study. Here we demonstrate discrepant findings regarding HR-SB being a risk factor for OPSCC.

**Abstract:**

(1) Background: Several lines of evidence established a link between high-risk (HR) sexual behavior (SB), the persistence of human papillomavirus (HPV) DNA in saliva, and the presence of oncogenic HR-HPV subtypes in oropharyngeal squamous cell carcinoma (OPSCC). A highly influential case-control study by D’Souza et al. comparing OPSCC patients and ENT patients with benign diseases (hospital controls) established HR-SB as a putative etiological risk factor for OPSCC. Aiming to replicate their findings in a nested case-control study of OPSCC patients and propensity score (PS)-matched unaffected controls from a large population-based German cohort study, we here demonstrate discrepant findings regarding HR-SB in OPSCC. (2) Methods: According to the main risk factors for HNSCC (age, sex, tobacco smoking, and alcohol consumption) PS-matched healthy controls invited from the population-based cohort study LIFE and HNSCC (including OPSCC) patients underwent interviews, using AUDIT and Fagerström, as well as questionnaires asking for SB categories as published. Afterwards, by newly calculating PSs for the same four risk factors, we matched each OPSCC patient with two healthy controls and compared responses utilizing chi-squared tests and logistic regression. (3) Results: The HNSCC patients and controls showed significant differences in sex distribution, chronologic age, tobacco-smoking history (pack years), and alcohol dependence (based on AUDIT score). However, PS-matching decreased the differences between OPSCC patients and controls substantially. Despite confirming that OPSCC patients were more likely to self-report their first sexual intercourse before age 18, we found no association between OPSCC and HR-SB, neither for practicing oral-sex, having an increased number of oral- or vaginal-sex partners, nor for having casual sex or having any sexually transmitted disease. (4) Conclusions: Our data, by showing a low prevalence of HR-SB in OPSCC patients, confirm findings from other European studies that differ substantially from North American case-control studies. HR-SB alone may not add excess risk for developing OPSCC.

## 1. Introduction

Human papillomavirus (HPV) is a driver of a subset of head and neck squamous cell carcinoma (HNSCC), in particular, HPV-driven oropharyngeal squamous cell carcinoma (OPSCC) emerging from epithelia lining Waldeyer’s ring, in particular. As high-risk oncogenic HPV subtypes such as HPV16 are transmitted via body fluids containing virus particles, and other HPV-driven cancers emerge from the epithelia of the uterine cervix and the anogenital region, HPV-driven OPSCC is considered a sexually transmitted disease (STD). A very influential case-control study by D’Souza et al. [1] compared 100 OPSCC and 200 age- and sex-matched patients from the same ENT hospital who were accrued as controls. This study established higher frequencies of antibodies to HPV16 proteins, and, in particular, to anti-HPV16 early proteins E6 and E7 as markers for HPV-driven OPSCC characterized by HPV-DNA positivity. To be more precise, 72% of tumors in their OPSCC sample were positive for HPV DNA, and the prevalence of antibodies to either E6 or E7 among the 100 OPSCC was 64% [1]. After earlier hints from other studies [2,3,4,5,6], this study was responsible for the definitive acceptance of “high risk sexual behavior” (HR-SB) as being causatively involved in the etiology of HPV-driven OPSCC, as 88% of OPSCC patients reported a lifetime number of ≥1 oral sex partners, accompanied by an increased prevalence of HPV16 or any HPV infection in the oral cavity of OPSCC patients (32% and 37% compared to 4% and 6% in controls) reflected by odds ratios (ORs) of 11.3 (5.0–25.7) and 10.0 (4.8–20.7) for OPSCC [1]. This study, however, investigated patients from a single tertiary American hospital and utilized statistical models and adjustment for the factors age, sex, tobacco, and alcohol, which are among those known to be causatively involved in the development of HNSCC (includingOPSCC), to elucidate the significance of HR-SB in this regard. As case-control studies and adjusting for confounders can be problematic in respect to achieving reliable statistical inference and relying solely on a single case-control study from another part of the world, the transferability of such findings might be limited through deviating prevalence and distribution of clinical characteristics, deviating socio-cultural environment, and differing covariates, as well as other unknowns, including the genetic background. For generalizability and drawing conclusions for the general population, the selection of cases and controls studied is critical. Substantial bias can be introduced whenever the cases and controls are not randomly chosen or are selected from a sample that is not representative for either the cases, controls, or both. Such results may rather reflect the special property of the particular sample than be representative of the general population, thus potentially leading to misinterpretations. Hence, even well-conducted case-control studies are at an increased risk for far-fetched extrapolation of their findings to unrelated populations. Preferable to case-control studies and adjusting for already known risk factors are population-based cohort studies. When executing a nested case-control study, utilizing healthy participants of a randomly drawn sample from the same cohort should result in healthy controls and superior controls compared to any kind of patients from the same hospital. This is obviously true, as “controls not affected by the disease” came to the hospital with a substantial medical need for treatment of another disease. Such controls inherently introduce substantial selection bias despite any statement that they had “benign disease not related to HNSCC”, as in the aforementioned study [1]. 

It remains questionable how such patients can be seen as “healthy controls”, as they are acceptable only in absence of any other “control”. However, most epidemiologic investigations in HNSCC report substantial differences between the general population and HNSCC patients, which are predominantly older males, and the general population regarding substantially higher prevalence of high-level alcohol consumption and tobacco-smoking history, and simultaneous exposure to both, in particular. This extends to a multitude of other occupational and environmental exposures. The impact of the most dominant risk factors could be responsible for underestimating the impact of any other causative factor, as they are confounders and introduce confounding bias. Adjusting for these confounders within a case-control study lowers the power, and matching in case-control studies introduces additional sources of bias, e.g., colliding bias [7]. However, matching within a cohort study removes both types of bias [7,8]. 

Drawing a propensity score (PS)-matched sample of participants attending the same cohort study based on the major risk factors for HNSCC (tobacco-smoking history, alcohol-consumption level, age, and sex) before assessing the distribution of covariates between OPSCC and PS-matched controls has the potential to reduce analytical bias further. As we had the unique opportunity to perform such an analysis in the framework of the LIFE study [9,10,11], we here demonstrate the discrepant findings respective to self-reported sexual behavior in a PS-matched nested case-control study of OPSCC patients and unaffected controls from a large population-based German cohort study.

## 2. Materials and Methods

### 2.1. Study Population and Patient Samples

The LIFE A1 Adult Study of the Leipzig Research Centre for Civilization Diseases (LIFE [9,10]) is a large population-based cohort study. Within LIFE, a total of 10,000 adults were scheduled to be recruited randomly from the City of Leipzig (cohort A1) to serve as a control sample for various diseases, including HNSCC. The sub-project LIFE B7 HNSCC [10,11] was a cohort study in the framework of the same population-based cohort study LIFE, and it was conducted according to the guidelines of the Declaration of Helsinki and approved by the Ethics Committee of the University Leipzig (votes 201-10-12072010 and 202-10-12072010). The LIFE study provided a rationale to identify a nested sample of *n* = 300 volunteers from the LIFE A1 Adult study to serve as controls for our cohort of HNSCC patients to allow for the identification of risk factors for HNSCC and to gain information about gene–environment interaction not only of well-known risk factors (alcohol and smoking) by additionally investigating other lifestyle factors potentially involved in HNSCC etiology, including sexual behavior. From 4 August 2010 to 18 July 2012, we enrolled a total of *n* = 450 patients who were suspected of having head and neck cancer. A cross-sectional comparison of HNSCC patients (cases from B7) and controls without HNSCC (from A1) was designed as a nested case-control study with 1:1 matching. We used the main risk factors for HNSCC—male sex, chronologic age, alcohol consumption, and tobacco smoking history (pack years smoked)—of the first 147 patients accrued to calculate PSs in B7 and all 6798 A1 participants at that time. With a scheduled sample size of 300, we consecutively invited A1 participants according to their PS, in descending order, to have the same interview used in B7 (Figure 1). According to weekly sent invitations, 303 out of the 698 highest-scoring A1 controls responded (response rate 43.4%), provided informed consent, and completed the interview to serve as a reference.

### 2.2. Matching Process and Demographic Variables

The first idea to match 147 HNSCC cases and 147 controls out of 6798 volunteers as “twins” via an exact matching due to the specified risk factors (tobacco smoking, alcohol consumption, age, and sex) failed. We found only 57 pairs among 147 HNSCC cases and 6798 LIFE A1 Adult volunteers with an identical characteristic according to sex, age, stratum of pack years (10 PY increments), and belonging to the same out of the four daily alcohol-consumption categories (0 or <1 g/d, 1–30 g/d, 31–60 g/d, or >60 g/d) of which the controls could have been invited for the second visit to record information about sexual behavior, etc. Moreover, these exact matching pairs were HNSCC patients and controls with predominantly low exposure to both tobacco and alcohol. These are risk-factor characteristics that are very common in A1 Adult participants but rarely found in HNSCC patients; hence, only 57 HNSCC cases not representative for HNSCC could be matched. 

As shown in Table 1, in our sample of HNSCC cases, 36% had >38 PY and were current smokers at the time of diagnosis. This exposure is mostly not reached by “normal” volunteers and is very seldom reported by healthy adults, including the random sample from our population-based cohort study LIFE A1 Adult. As high-level daily alcohol consumption is the second most common among risk factors for HNSCC and is consistently identified in the multitude of epidemiological studies, and, moreover, a simultaneously high exposure to alcohol and tobacco smoking is found in HNSCC patients, this was the rationale to match HNSCC cases and controls according to these risk factors, as well as to the other major risk factors for HNSCC, age, and (male) sex.

Therefore, we decided to perform PS-matching and used the list generated and sorted according to the highest PS, which was predominantly linked to the highest tobacco smoke category, to invite potential A1 Adult participants to participate in the nested case-control study and come for a second visit. As inviting *n* = 698 A1 Adult volunteers was required to accrue 303 controls who would attend the second visit and answer our questionnaires, the presence of high-level tobacco smoking accompanied by a high level of alcohol exposure remained lower in responding and interviewed controls. In the sample of 303 controls, a higher exposure to alcohol (but often without smoking) could be observed that had (at least partially) compensated for the often-lower level of smoking of the individual and increased its PS and allowed for inviting the A1 Adult volunteer to serve as a control. 

Among the final sample of *n* = 317 HNSCC patients who answered >50% of the questionnaire items and *n* = 303 A1 Adult controls accepting the invitation to participate as controls for the nested cohort study, we applied a second round of PS matching of all *n* = 112 OPSCC patients (patients with the primary tumor in the base of the tongue = ICD-10-C01, the uvula = C05, the palatine tonsils = C09, or other parts of the oropharynx = C10) and the *n* = 303 A1 Adult volunteers. Finally, we obtained 94 OPSCC patients matched with 188 PS-matched controls (Table 2). According to this enrollment strategy, only 14 women were suitable as controls, but obviously some male controls were equivalent or allowed data-driven PS-matching to female OPSCC patients. Overall, making use of PS-matching was able to reduce the otherwise larger distance between OPSCC cases and controls that can be described as being of a small effect. This can be concluded from the PS in *n* = 188 PS-matched controls (mean, 0.3137; standard deviation, 0.1650) vs. *n* = 94 PS-matched OPSCC (0.3778, 0.1510; *p* = 0.0017; *Cohen’s d* = 0.3993), vs. *n* = 66 p16+ OPSCC (0.3903, 0.1493; *p* = 0.0010; *Cohen’s d* = 0.4753), and vs. *n* = 33 HPV-driven OPSCC (0.3616, 0.1407; *p* = 0.1182; *Cohen’s d* = 0.2961). 

Clinical characteristics and demographic variables of cases and controls are provided for all participants in Table 1 and for the PS-matched subsample in Table 2. 

### 2.3. Questionnaires

We used the well-established questionnaires AUDIT [14] and Fagerström [15] to assess alcohol dependency and nicotine dependency, respectively. Besides interpretation of the obtained answers, score points handled as numerical values were also analyzed. Referring to D’Souza’s study [1], and with the aim of confirming their findings in a German cohort, we used the same cutoff values reported in their study to ask for lifetime numbers of sex partners. Due to low numbers of patients and controls reporting anal sex and same-sex partners in their study, we omitted asking the respective questions in our study. 

The items of the questionnaires were provided by a trained interviewer (a certified study nurse) who read out questions to the probands during a structured face-to-face interview. The probands were asked if they belonged to one of the predefined answer categories then recorded by the interviewer. 

### 2.4. Statistical Analyses and Calculation of Propensity Scores

The statistical analyses were performed using SPSS version 27 (IBM Corporation, Armonk, NY, USA) and included *Pearson’s* Chi-square (*χ*^2^) tests to assess differences between categorical variables, as well as logistic regression for multivariate analyses and the calculation of PS for PS-matching.

The PS calculation was performed based on the covariates smoking (in pack years, continuous), alcohol per day (categorical), chronologic age (continuous), and sex (categorical) automatically. This procedure runs a logistic regression on the group indicator and the covariates (predictors) and then uses the resulting PS (a value between 0 and 1) with the defined matching tolerance (caliper width of 0.1 was chosen, as this is recommended as the optimum compromise by most investigators) to select controls for cases. We used (by checking the respective checkboxes) the option to give priority to exact matches and to randomize the case order when drawing matches without resampling.

### 2.5. Molecular Analyses of HPV

The HPV DNA status and genotype were determined in 100 ng DNA of each sample as previously described [11]. RNA samples of HNSCC positive for the subtype HPV16 underwent analysis of E6*I transcripts by RT-PCR and were concluded to be positive for HPV16 RNA whenever HPV16 E6*I transcripts were detected. The CINtec kit (Roche) was used for the detection of p16 in formalin-fixed, paraffin-embedded primary tumor samples from OPSCC only. The detection of at least 20% stained tumor cells was used to conclude p16 positivity. HPV-related OPSCC was defined as p16 positivity, whereas the status HPV-driven was concluded only if OPSCC had simultaneous positivity for high-risk HPV DNA and/or RNA plus p16-positivity above the cutoff level of ≥70% OPSCC cells [16]. 

## 3. Results

From 4 August 2010 to 18 July 2012, *n =* 450 patients suspected of having head and neck cancer provided written informed consent and were accrued for LIFE B7. Patients without any sign of malignancy (*n* = 61) and those with synchronous or metachronous malignancy of other histology (*n* = 35) were excluded (Figure 1). Out of *n =* 354 potentially eligible patients with tumor sites in the head and neck region and pathologically confirmed squamous cell carcinoma without any other synchronous malignancy, 329 (93%) were enrolled and agreed to participate in the interview. Of these 329 HNSCC patients, 317 provided answers to more than 50% of questionnaire items (Figure 1). Among these *n =* 317 HNSCC were *n =* 112 OPSCC (ICD-10 codes C01, C05, C09, or C10). The characteristics of 317 HNSCC cases and 303 controls are shown in Table 1. 

Despite the PS-matched invitation of potential controls based on the main risk factors age, sex, tobacco smoking (expressed in pack years), and alcohol consumption, substantial differences were noticed. A history of heavy tobacco use and greater nicotine dependence (*Fagerström* questionnaire [15]) were more prevalent in those with HNSCC, whereas the controls had greater daily alcohol consumption, were older, and were more frequently of the male sex. The *AUDIT* [14] scores for alcohol dependence did not differ significantly between cases and controls. Probably related to the selection process of controls from the LIFE A1 Adult study, a history of former drinking was found only among HNSCC patients, leading to significant differences in this respect. 

The self-reported employment status did not differ significantly. Nevertheless, controls more often reported living in larger apartments with more rooms. HNSCC patients more often reported complete tooth loss and various aspects of impaired oral hygiene. 

Due to highly different characteristics in HNSCC and controls potentially hindering reliable comparisons, we performed PS matching of 112 OPSCC patients and 303 controls. Applying a caliper width of 0.1, we randomly assigned two controls to each OPSCC patient. This provided a total sample of 188 controls for 94 PS-matched OPSCC patients. The remaining 18 of the 112 OPSCC patients (16%) and 115 controls (38%) without compatible matching partners were excluded from further analysis (Figure 1 and Table 2). 

Out of the 94 OPSCC, 66 (70.2%) were deemed HPV-related, as they expressed p16. According to Table 2, the members of the PS-matched OPSCC subgroup and their PS-matched controls demonstrated a comparable distribution of chronologic age, average quantity of pack years, and alcohol dependence (based on *AUDIT* score). However, related to the selection process of controls, and even by using the specified caliper width of 0.1, some significant differences in the distribution of predictors used in the propensity-score-based automatic matching persist, as were described. Comparing 94 OPSCC cases and 188 PS-matched controls (Table 2) revealed narrower characteristics but also a higher exposure to tobacco (pack years smoked) and greater nicotine dependence (*Fagerström* questionnaire [15]) in OPSCC cases, whereas the controls had greater daily alcohol consumption, were older, and were more frequency of the male sex. Regarding the propensity scores, a high level for one risk factor in the absence of the other obviously could (partially) compensate for the other and resulted in a comparable PS despite deviating in pack-years tobacco-smoking history and/or daily alcohol consumption (Table 2). 

Within the PS-matched analysis, we found OPSCC patients to be more likely to self-report their first sexual intercourse before age 18. There were no differences in frequency of having ever had casual sex or using a condom usually or always. Sexually transmitted diseases (STDs) were more frequent among controls (not significant). However, the appearance of oral or genital warts and a positive family history of tumor disease or SCC was low but numerically slightly higher in OPSCC patients. We found no association between OPSCC and sexual behavior, neither for the numbers of oral-sex partners or vaginal-sex partners, as the lifetime numbers were significantly lower in OPSCC patients. This observation relates to all categories and not only to the most extreme numbers. However, there were no differences after the Bonferroni correction (Table 2). 

Venn diagrams for the distribution of ≥6 oral-sex partners and >25 vaginal-sex partners in 188 controls vs. either 94 OPSCC or 66 p16+ OPSCC demonstrate a rather reduced lifetime prevalence. Any unusual clustering or double-positive patterns were absent (Figure 2). The frequency of females among OPSCC and p16+ OPSCC was 18/94 (19.1%) and 12/66 (18.2%), respectively, and hence nearly identical. There was no significant difference between cases and controls regarding ≥6 oral sex partners; none of the 18 female OPSCC patients but 4 of 14 (28.6%) of the female controls reported a lifetime prevalence of ≥6 oral-sex partners (*p* = 0.015). Within the subgroup of 33 patients with HPV-driven (i.e., ≥ 70% p16+ HR-HPV DNA+) OPSCC, a comparable distribution regarding the number of oral- or vaginal-sex partners and a low frequency of HR-SB were observed, and especially among the 30 patients p16+ HPV16 DNA+ RNA+ OPSCC, as only three male patients (one patient each (3.3%)) were in the HR-SB groups reporting either a lifetime number of ≥26 vaginal-sex partners, ≥6 oral-sex partners, or both. Overall, HR-SB was lower in OPSCC patients than in controls, and this frequency was not increased in patients with HPV16-driven OPSCC cases. 

Having had any oral sex, in contrast, was not different in females (*p* = 0.265) but in males, with about a 19% higher proportion in controls (35.5% vs. 54.6%, *p* = 0.005). The logistic regression failed to demonstrate any link between the self-reported number of vaginal-sex partners, casual sex, or sexually transmitted disease and being diagnosed with OPSCC or p16+ OPSCC, in particular, whereas oral sex was found to be significantly protective for OPSCC according to an OR of 0.384 (95% CI, 0.206–0.716), *p* = 0.003, for 1–5; and an OR of 0.296 (95% CI, 0.113–0.770), *p* = 0.013, for ≥6 lifetime oral sex partners. However, the logistic regression demonstrated an earlier sexual debut (age < 18 compared to ≥ 18 years) in OPSCC patients being accompanied by an OR of 1.994 (95% CI, 1.054–3.773), *p* = 0.034, confirming the link between earlier sexual debut (age < 18 years).

## 4. Discussion

Circumventing some potential sources of bias related to case-control studies, we executed a nested case-control study utilizing consecutive accrued OPSCC cases and PS-matched controls from the German population-based cohort study LIFE [9,10] to answer the question of whether HR-SB—oral sex, in particular—is a relevant etiologic factor in the development of HPV-related OPSCC. Our study did not show huge differences in sexual behavior between OPSCC patients and the controls or a comparable prevalence of self-reported characteristics, including HR-SB, thus demonstrating a replication failure of the findings from the American case-control studies, especially the most often cited study of D’Souza et al. [1]. 

Generally, in case-control studies, newly diseased individuals (cases) are compared with non-diseased individuals (controls) regarding various risk factors (exposure, e.g., age, sex, tobacco smoking, and alcohol drinking), preferably those with an already-known impact. Such controls can either be from the same population or so-called hospital controls, i.e., *control patients* attending the same hospital as the cases but due to another disease. Superior regarding representativeness is a random sample of the same population (population-based controls, such as LIFE A1 Adult).

In either case, the selection of controls must be performed with great care to avoid selection bias [17,18,19]. The controls should always be recruited from the same reference population from which the studied case group is derived. In other words, considering time and place, an individual should be included as a control in the study only if, assuming that he or she would have developed the disease, he or she would also have been eligible for the case group (“study base principle”). Furthermore, controls should be selected randomly in a way to minimize the risk of uncontrolled confounding (“deconfounding principle”).

Referring to these principles, we note that D’Souza [1] selected controls from a set of patients with benign diseases who were accrued in the same hospital during the recruitment period, after the patients were referred to the hospital and accrued from a larger OPSCC sample. Therefore, neither the cases nor the controls are representative for all OPSCC or represent a randomly recruited healthy population. These controls were matched for age and sex, thus reducing the confounding bias by introducing colliding bias [7,8]. However, the cases differed substantially from the controls in other risk factors for developing OPSCC, such as heavier tobacco smoking, alcohol consumption, and the use of marijuana [1], and it remained unclear if these differences are the same in European samples with deviating distribution in a number of risk factors. Indeed, one central question in case-control studies is whether and in which form the control group is defined and how reliable controls should be selected and eventually matched to cases regarding known confounders and their distribution to increase the possibility to elucidate risk factors other than those well-established risk factors. In practice, such matching is achieved by different forms of matching [19]; in our study, this was achieved by inviting real healthy controls who were unaffected by the disease (HNSCC) out of a population-based cohort study [9,10], according to their PSs, and thereafter matching each OPSCC with two controls according to the four main risk factors for HNSCC, using newly calculated PSs. Notably, these patients are truly representative for all OPSCC cases treated in our hospital, which treats the majority of HNSCC patients living in the Leipzig region, congruent with the LIFE study area [9,10].

Patients who are included in case-control studies need to be truly representative for all cases of a given population. Therefore, specialized centers or tertiary hospitals with selective referral may not be that representative for both OPSCC cases and healthy controls, as intended. Indeed, Maura Gillison [20], in response to a letter questioning the representativeness of their findings [1], stated that the authors “cannot exclude the possibility that subjects who did not have traditional risk factors were more likely to participate in [their] study as it was performed in a hospital and was not population-based”. Unfortunately, the scientific community mostly neglected this information, thus limiting the transferability of the findings to the general population and the interpretation of the correlation between HR-SB and OPSCC in their study [1]; instead, they interpreted the data from their study as evidence for a causative involvement of HR-SB in the etiology of OPSCC in general. However, there might be numerous differences between OPSCC patients in various regions of the world. Moreover, even the best study performed in a single hospital might not result in findings that are applicable to all other populations in the world or provide evidence for a causative involvement of HR-SB in general. Their well-conducted study showed a correlation between antibodies to HPV early proteins and HPV-positive OPSCC patients who had also had a higher prevalence of HR-SB. The latter statistical association observed unfortunately cannot explain how differences in HR-SB translate into HPV-related OPSCC or if other etiologic factors are more important. Focusing on comparisons of cases and controls derived from the same population and largely identical distribution of known major risk factors other than the covariate analyzed would have been required to draw such conclusions. 

Indeed, PS-matching decreased the differences between HNSCCs, including OPSCC, patients, and controls, in our study, despite appearing to be unable to completely eliminate all confounding bias, as more men than women and a higher level of alcohol consumption within controls were found, whereas the cases included more smokers in higher tobacco-smoking categories, and some other differences were also observed (Table 1 and Table 2). 

However, there were no differences in the self-reported numbers of lifetime vaginal-sex and oral-sex partners between our PS-matched sample of OPSCC patients and controls. Comparing our findings to the Study of D’Souza et al. [1], we observed a substantially lower prevalence of HR-SB in controls and an even lower prevalence of HR-SB in OPSCC cases (Table 3 and Table 4), arguing against a substantial impact of HR-SB and a large fraction of OPSCC that is attributable to HR-SB. The consistent absence of HR-SB in the overwhelming majority of HPV-driven OPSCCs stands against the argument of a lowered frequency of HPV-driven OPSCC in our cohort (35.1% vs. 64% in their sample, according to seropositivity for HPV16 E6 and/or E7 antibodies) that would have lowered the chance to detect an impact of HR-SB on the development of HPV-driven OPSCC.

Only 37.2% of LIFE B7 OPSCC patients stated ever having practiced oral sex compared to 88% of their cases [1]. Likewise, the lifetime-numbers of oral and vaginal sex partners they reported were much higher. Differences might be related to deviating sexual behavior norms in Germany versus North America [1], as practicing oral–genital sex is generally reported more often in American [1,2,3,4,5,6,21] than in European studies [22,23,24,25,26,27].

There are a number of studies highlighting either a lower age at sexual debut/first intercourse or a higher proportion of OPSCC patients reporting an age below 18 years at first intercourse, and our study confirms these reports [1,3,27,28]. However, this behavioral aspect is often discussed as being linked to the rather low income of parents [29]. However, we cannot exclude that a lower age at first intercourse increases the risk for OPSCC via the increased vulnerability of the epithelia of younger persons, making them more susceptible to becoming infected with HPV. 

Regarding other European trials, Tachezy et al. [23] collected the data of 86 patients with a primary cancer of the oral cavity or oropharynx and 124 controls in the Czech Republic, regarding demographics, behavioral risk factors, and risks related to HPV exposure. Referring to sexual behavior, data with an increased risk for HNSCC could be shown after adjusting for age and the consumption of alcohol and tobacco only for practicing oral–anal sex (OR 4.3, 95% CI 1.3–14.8, *p* = 0.02). The number of sex partners (<6 vs. >6; OR 1.2, 95% CI 0.6–2.7, *p* = 0.59), as well as practicing oral–genital sex (OR 0.5, 95% 0.2–1.2, *p* = 0.14) could not be confirmed as risk factors for HNSCC and also not for OPSCC [23]. Other European studies also failed to demonstrate any significant correlation between HR-SB and oral or oropharyngeal cancer [22,24,25].

Farsi et al. [30] published a meta-analysis of 20 case-control studies of sexual behavior in patients with HNSCC. Among all included studies, nine were from North America, five from Europe, four from Latin America, one from Asia, and one from Oceania. Only three of them adjusted their analyses for HPV status. According to ORs in random-effect models including all studies, an increased risk of HNSCC was found for the number of sexual partners (19 studies; OR 1.29, 95% CI 1.02–1.63) and the number of oral-sex partners (5 studies; OR 1.69, 95% CI 1.00–2.84). After excluding studies contributing the most to heterogeneity (e.g., D’Souza et al. [1]) or not adjusting for age, sex, smoking, and alcohol consumption, neither the number of sexual partners nor practicing oral sex (OR 0.95, 95% CI 0.75–1.20 and 1.03, 95% CI 0.84–1.26, respectively) was associated with oral or oropharyngeal SCC. This is in line with our findings. They concluded that observed associations might be partly attributed to confounding [30], and our findings support their interpretation.

However, the prevalence of HPV-related OPSCC per se was found to be significantly higher in the geographic region of North America than Europe [31,32]. Similar to the HPV prevalence of 72% in OPSCC in D’Souza’s study [1], prevailing estimates indicate that, in the United States, approximately 60% to 70% of OPSCCs are caused by (or at least are related to) HPV infection [21,33,34]. Similarly, in a comprehensive review of the global burden of infection-associated cancers, systematic reviews estimated that HPV infection accounted for 56% to 60% of OPSCCs in North America, compared with 17% to 41% of OPSCCs in European countries [31,35]. As HPV infection itself is associated with HR-SB, it serves as a confounder in many of the studies presented. HPV-induced oropharyngeal carcinomas have steadily increased in recent decades, possibly due to changing sexual behavior of the population [36,37]. However, not every person with oropharyngeal HPV infection or a risky lifestyle develops HNSCC. Most HPV infections are cleared by the immune system, eventually leading to humoral and cellular immunity; the presence of antibodies to HPV capsid proteins and L1, in particular, may be indicators for prior infection. Indeed, the frequency of antibodies to L1, but not to the oncogenic E6 and E7 proteins, demonstrates an association with self-reported sexual behavior [38]. Within HPV subtypes, e.g., HPV16, there are variants with varying distributions in geographical regions. While the European HPV16 prototype (E) is dominant in Germany, the HPV16 Asian-American (AA) variant is prevalent in the United States of America. This could be important, as HPV16 AA is more oncogenic [39,40,41]. Regional differences in genetic or immunological predisposition for a non-cured HPV infection may also lead to an increased frequency of HPV-positive OPSCC in different geographic regions. The risk for cancer evolvement is increased by genetic variants in genes encoding enzymes involved in DNA repair or the metabolism of alcohol [42,43]. They also increase the risk independent of lifestyle-associated risk factors [43]. Genetic instability and carcinogen exposure lead to somatic mutations and an increased tumor mutational burden if they cannot be controlled by the immune system. Higher cancer incidences are linked to immune defects in the natural killer cell (NK cell) and T-cell system. As T-cell responses to peptides of oncogenes critically depend on binding and proper presentation [44] by human leukocyte antigen (HLA) antigens and their combinations (haplotypes), and the distribution of these differs according to ancestry, the genetic background of patients could also be involved in deviating findings by comparing diverse populations. 

HPV-driven cancers such as cancer of the uterine cervix and vulvar, penile, anal, and oropharyngeal cancer are found to be increased in immune-suppressed populations [5], either iatrogenic induced after organ transplantation or related to immune deficiency. Immune deficiency can be inherited or acquired [45,46]. It is well-known that increased age, marijuana exposure, or HIV infection are linked to immune deficiency, contributing to an elevated frequency of opportunistic infections and particular cancers in the affected. Marijuana use and Human Immunodeficiency Virus (HIV) seropositivity are linked to reduced clearance of HPV and persistence of oral HPV infection [5]. Thus, regional variation in the co-incidence of undetected or untreated HIV infection also affects the likelihood of HPV-induced cancer—OPSCC, in particular. In the absence of (uncontrolled) HIV infection or persistent marijuana use, age-related immune suppression and age-related loss in immune surveillance remain the essential contributors to increased cancer risk even in prior immune-competent subjects. This might be the case in our cohort of HNSCC patients without any HIV-positive case and without marijuana use and a higher age compared to other studies from the *U.S*. [1,6]. This might also be reflected by the lower prevalence of HPV-driven OPSCC in Germany and especially in our cohort.

We could not confirm the numbers of oral- and vaginal-sex partners, as well as ever having practiced oral–genital sex, as risk factors for HNSCC or even p16+ OPSCC (Table 2) and, among them, the HPV-driven cases (Figure 2). The significantly increased proportion of OPSCC patients who were aged <18 years at the time of first intercourse was the only finding that was consistent with many studies, including those from the *U.S.* [1,3,27,28]. This may argue for a so-far mechanistically unexplained risk for oncogenic infections linked to increased vulnerability at a younger age. However, an earlier sexual debut correlates only with an increased prevalence of antibodies to L1 but not E6 or other early proteins [38,47]. Therefore, most early vaccinations of girls and boys would probably be advantageous [42,48]. In Germany, the *Standing Committee on Vaccination* (STIKO) began to recommend HPV vaccination for young females aged 12 to 17 in 2007. Only since 2018 has the HPV vaccination been recommended for young men (and women) aged 9 to 14. Considering that our study was conducted during the years 2010 to 2012 and regarding the age distribution in our study with only one HNSCC case at age 25, one case at age 35, one control at age 36 but no control in the age category of 18 to 30, we expect no impact of HPV vaccination on HR-SB, immune competence towards HPV, or occurrence of neoplastic transformation of epithelia by HPV in our cohort. Further studies in years to come should explore the potential effects of HPV vaccination on the occurrence of OPSCC.

The limitations of our study arise from using a modified questionnaire that explicitly asks for categories according to the cutoff values for the number of sexual partners, as in D’Souza et al. [1], and not allowing free responses. Moreover, we are aware of the potential for recall bias or misreporting because the interview, as opposed to a self-administered questionnaire, could also have led to inaccurate responses when providing such sensitive information due to social suitability or shame in a face-to-face setting. This could have contributed to fewer subjects reporting HR-SB in both the cases and controls. In addition, we had only sufficiently high case numbers according to p16 positivity or negativity of the tumor material but not according to HPV seropositivity of probands. Moreover, we detected only 35.2% of HPV-driven OPSCC among the 94 PS-matched OPSCC cases. Given the low prevalence of HPV-driven OPSCC, and even more of HR-SB in patients with OPSCC, p16-positive OPSCC, and even HPV-driven OPSCC, it is reasonable to believe that the low numbers might have led to a substantially underestimated effect, or that we may have missed an impact of HR-SB on development of a larger proportion of OPSCC patients. 

Our study contributes controversial data to the discussion about the association between HR-SB and OPSCC, and further studies are needed as substantial differences between populations exist, e.g., European and American. HPV-driven or p16+ OPSCC in Leipzig, Germany, and Baltimore, Maryland, appear to have not only different prevalences but also at least partially deviating characteristics, thus altogether impairing the transferability of the findings between geographical regions. We hereby suggest paying attention to an appropriate interpretation of data from various regions and obtaining data from studies performed in comparable populations that avoid bias and the misinterpretation of results. The appropriate selection of cases and matched healthy controls from population-based cohort studies appear to be a requirement for conducting insightful case-control studies. 

## 5. Conclusions

The data we collected on the sexual behavior of OPSCC patients and controls are similar to those of other European studies but differ substantially from those of North American studies. The differences may be due to different regional–cultural sexual behaviors or linked to particular HPV variants. A varying genetic or immunological predisposition may also cause a selective immune incompetence or more general immune deficiency, allowing an oral HPV infection to evolve into cancer. Consequently, and according to the low prevalence in our OPSCC patients, sexual behavior alone may not be the sole or most responsible contributor to the development of OPSCC.

## Figures and Tables

**Figure 1 cancers-15-03356-f001:**
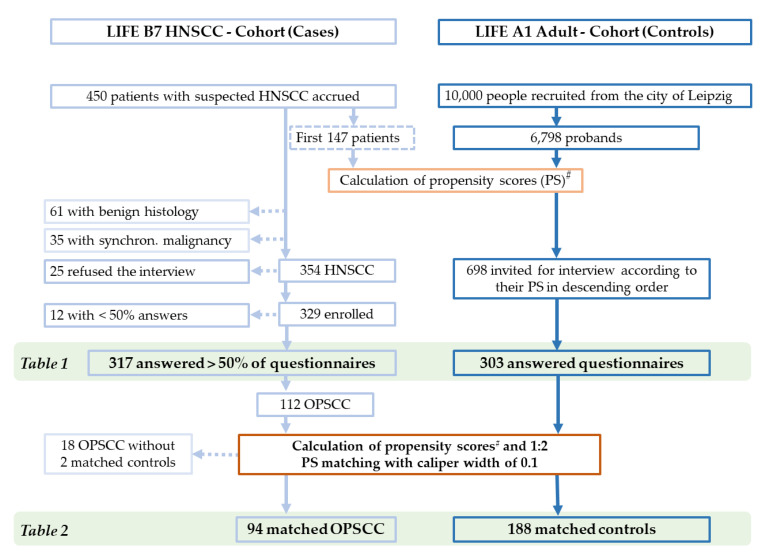
CONSORT diagram, showing the selection process of cases (HNSCC and OPSCC) and controls for comparisons. Selected cohorts shown in tables are marked. ^#^ Calculation of PS based on the main risk factors for HNSCC: male sex, chronologic age, alcohol consumption, and tobacco-smoking history (pack years smoked).

**Figure 2 cancers-15-03356-f002:**
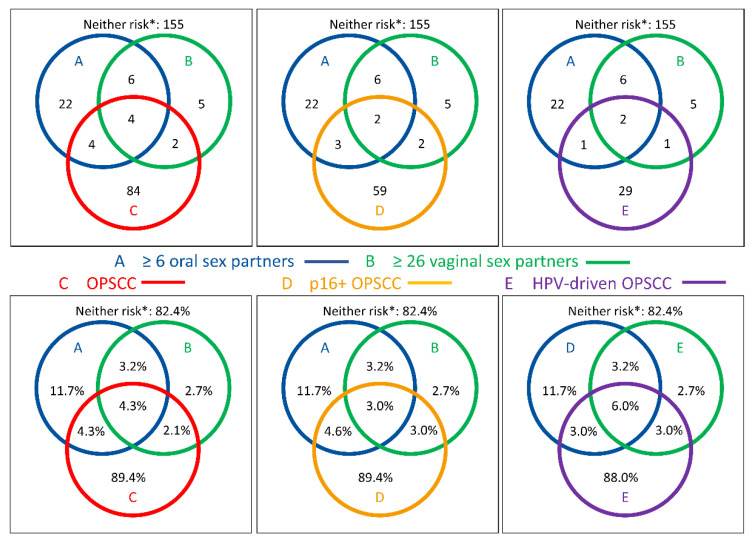
Venn diagrams showing the distribution of self-reported high-risk sexual behavior (HR-SB) among *N* = 188 propensity-score-matched healthy adults and among *N* = 94 oropharyngeal squamous cell carcinoma (OPSCC, C; red circles, left) patients, the subgroup of *n* = 66 p16-positive OPSCC patients (D, orange circles, middle), and *n* = 33 HPV-driven (p16+ ≥70%, HR-HPV DNA+) OPSCC patients (E, purple circles, right) according to numbers (upper row) and percentage (lower row) regarding the number of oral-sex partners ≥ 6 (blue) and number of vaginal-sex partners ≥ 26 (green). * Neither risk depicts PS-matched controls without HR-SB and OPSCC. Please note that there is no adjustment of diameters for proportionality.

**Table 1 cancers-15-03356-t001:** Explanatory variables for patients with head and neck squamous cell carcinoma (HNSCC) and adult controls invited according to their propensity score (PS). Significant *p* values are bold.

	Total		LIFE B7 HNSCC Patient		LIFE A1 Adult Control			
	*n* (%)		*n* (%)		*n* (%)		OR (95% CI)	*p-*Value ^a^
Demographic characteristics								
Sex								
Female	62	(10.0)	48	(15.1)	14	(4.6)	3.683 (1.985–6.834)	**<0.0001**
Male	558	(90.0)	269	(84.9)	289	(95.4)		***0.0022*** ***^B^***
Age Score								
18–50 years	87	(14.0)	55	(17.4)	32	(10.6)	Ref. (0.540–1.852)	**<0.0001**
51–60 years	180	(29.0)	118	(37.2)	62	(20.5)	1.107 (0.650–1.888)	***<0.0001*** ***^B^***
61–70 years	184	(29.7)	84	(26.5)	100	(33.0)	0.489 (0.290–0.825)	
>70 years	169	(27.3)	60	(18.9)	109	(36.0)	0.320 (0.187–0.548)	
Smoking								
Never smoker	159	(25.6)	50	(15.8)	109	(36.0)	Ref. (0.623–1.606)	**<0.0001**
Former Smoker	174	(28.1)	42	(13.2)	132	(43.6)	0.694 (0.428–1.124)	***<0.0001*** ***^B^***
Current Smoker	287	(46.3)	225	(71.0)	62	(20.5)	7.911 (5.110–12.25)	
Pack years (Quartiles)								
0 PY	159	(25.6)	50	(15.8)	109	(36.0)	Ref. (0.623–1.606)	**<0.0001**
>0–20 PY	165	(26.6)	68	(21.5)	97	(32.0)	1.528 (0.968–2.412)	***<0.0001*** ***^B^***
21–38 PY	140	(22.6)	85	(26.8)	55	(18.2)	3.369 (2.092–5.426)	
>38 PY	156	(25.2)	114	(36.0)	42	(13.9)	5.917 (3.636–9.630)	
Pack Years ^b^								
Mean (95% CI)			30.24	(27.68–32.81)	17.29	(14.49–20.08)		**<0.0001 ^b^**
Fagerström Test for Nicotine Dependence (Score)								
Low (0–2)	439	(72.7)	176	(56.8)	263	(89.5)	Ref. (0.763–1.310)	**<0.0001**
Low to Moderate (3–4)	70	(11.6)	59	(19.0)	11	(3.7)	8.015 (4.096–15.68)	***<0.0001*** ***^B^***
Moderate (5–7)	81	(13.4)	65	(21.0)	16	(5.4)	6.071 (3.401–10.84)	
High (8–9)	14	(2.3)	10	(3.2)	4	(1.4)	3.736 (1.154–12.10)	
Fagerström Test ^b^								
Mean (95% CI)			2.37	(2.10–2.65)	0.65	(0.45–0.86)		**<0.0001 ^b^**
Alcohol Use								
Never drinker	69	(11.1)	34	(10.7)	35	(11.6)	Ref. (0.513–1.949)	**<0.0001**
Former Drinker	42	(6.8)	42	(13.2)	0	(0)	87.46 (5.176–1478.0) ^c^	***<0.0001*** ***^B^***
Current Drinker	509	(82.1)	241	(76.0)	268	(88.4)	0.926 (0.560–1.531)	
Daily Amount of Alcohol								
0 or <1 g/d	69	(11.1)	34	(10.7)	35	(11.6)	Ref. (0.513–1.949)	**<0.0001**
1–30 g/d	184	(29.7)	110	(34.7)	74	(24.4)	1.530 (0.877–2.669)	***<0.0001*** ***^B^***
31–60 g/d	110	(17.7)	78	(24.6)	32	(10.6)	2.509 (1.342–4.693)	
>60 g/d	257	(41.5)	95	(30.0)	162	(53.5)	0.604 (0.353–1.031)	
AUDIT Test for Alcohol Dependence (Score)								
High Dependence (≥8)	136	(21.9)	71	(22.4)	65	(21.5)	1.057 (0.722–1.546)	0.7761
Low Dependence (<8)	484	(78.1)	246	(77.6)	238	(78.5)		*>0.9999* *^B^*
AUDIT Test ^b^								
Mean (95% CI)			5.08	(4.53–5.64)	5.23	(4.76–5.70)		0.6923 ^b^
Age at first employment ^b^								
Mean (95% CI)			16.82	(16.33–17.30)	16.88	(16.54–17.23)		0.8306 ^b^
Years in employment ^b^								
Mean (95% CI)			38.04	(36.93–39.16)	37.31	(36.16–38.45)		0.3656 ^b^
Quantity of jobs ^b^								
Mean (95% CI)			4.29	(3.97–4.60)	4.18	(3.89–4.47)		0.6149 ^b^
Number of rooms in dwelling								
1	27	(4.5)	23	(7.5)	4	(1.3)	Ref. (0.223–4.489)	**<0.0001**
2	63	(10.4)	51	(16.7)	12	(4.0)	0.739 (0.215–2.539)	***<0.0001*** ***^B^***
3	166	(27.5)	83	(27.1)	83	(27.9)	0.174 (0.058–0.525)	
4	176	(29.1)	67	(21.9)	109	(36.6)	0.107 (0.035–0.323)	
5	172	(28.5)	82	(26.8)	90	(30.2)	0.158 (0.053–0.478)	
Square meters of living space								
<50 m^2^	80	(13.2)	63	(20.7)	17	(5.7)	Ref. (0.469–2.133)	**<0.0001**
50–70 m^2^	192	(31.8)	103	(33.8)	89	(29.8)	0.312 (0.170–0.573)	***<0.0001*** ***^B^***
70–100 m^2^	199	(32.9)	79	(25.9)	120	(40.1)	0.178 (0.097–0.326)	
100–130 m^2^	95	(15.7)	41	(13.4)	54	(18.1)	0.205 (0.105–0.401)	
>130 m^2^	38	(6.3)	19	(6.2)	19	(6.4)	0.270 (0.117–0.620)	
Daily tooth brushing								
No	27	(4.4)	27	(8.7)	0	(0)	57.72 (3.504–950.7) ^c^	**<0.0001**
Yes	580	(95.6)	283	(91.3)	297	(100)		***<0.0001*** ***^B^***
Tooth Loss								
None	34	(5.6)	11	(3.5)	23	(7.7)	Ref. (0.362–2.762)	**<0.0001**
Some	335	(55.0)	129	(41.3)	206	(69.4)	1.309 (0.618–2.776)	***<0.0001*** ***^B^***
Many or Complete	240	(39.4)	172	(55.1)	68	(22.9)	5.289 (2.445–11.44)	
Mouthwash during past year								
<1 time/day	303	(59.6)	151	(63.2)	152	(56.5)	Ref. (0.727–1.375)	**0.0067**
1–2 times/day	191	(37.6)	77	(32.2)	114	(42.4)	0.680 (0.471–0.981)	*>0.9999* *^B^*
≥3 times/day	14	(2.8)	11	(4.6)	3	(1.1)	3.691 (1.010–13.49)	
Tumor site								
Oropharynx			112	(35.3)	-	-		
Oral			68	(21.5)	-	-		
Hypopharynx			42	(13.2)	-	-		
Larynx			80	(25.2)	-	-		
NSCCUP ^d^			9	(2.8)	-	-		
Other			5	(1.6)	-	-		

^a^ The *p*-values are from Pearson’s Chi-square (χ^2^) test. ^B^ Bonferroni-corrected *p*-values for multiple comparisons italic. ^b^ The *p*-value is from the heteroscedastic *t* test. ^c^ Odds ratio adjusted according to Cox and Moses by adding 0.5 to each cell to reduce bias and prevent division by zero caused by empty cells [12,13]. ^d^ Neck squamous cell carcinoma from an unknown primary tumor. Please note that due to rounding errors, the percentages shown may not sum up to 100 percent, as the distribution is shown only for a percentage of the available data.

**Table 2 cancers-15-03356-t002:** Explanatory variables and sexual behavior among 94 OPSCC patients and 188 propensity score (PS)-matched controls from the cohort study LIFE that could be randomly assigned to an individual OPSCC patient in a 2:1 ratio based on a caliper width of 0.1. ^#^ Significant *p* values are bold.

	Total		LIFE B7 OPSCC Patient		PS-Matched Control			
	*n* (%)		*n* (%)		*n (*%)		OR (95% CI)	*p*-Value ^a^
p16-Positivity								
p16 positive (HPV-related)			66	(70.2)	*-*	-		
p16 negative			28	(29.8)	*-*	-		
HPV status								
HPV-DNA neg p16 neg			23	(24.5)				
HPV16-DNA+ p16 neg			5	(5.3)	-	-		
HPV-DNA neg p16+ (≥20%)			33	(35.1)	-	-		
HPV-driven p16+ ≥70% and			33	(35.1)	-	-		
* HPV16 DNA+ E6*I RNA+*			*30*	*(31.9)*	*-*	*-*		
*HPV18 DNA+*			*1*	*(1.1)*	*-*	*-*		
*HPV39 DNA+*			*1*	*(1.1)*	*-*	*-*		
*HPV51 DNA+*			*1*	*(1.1)*	*-*	*-*		
Demographic characteristics								
Sex								
Female	32	(11.3)	18	(19.1)	14	(7.4)	2.944 (1.392–6.223)	**0.0035**
Male	250	(88.7)	76	(80.9)	174	(92.6)		*0.5972 ^B^*
Age ^b^								
Mean (95% CI)			59.43	(57.52–61.34)	60.37	(59.06–61.68)		0.4240 ^b^
Age Score								
18–50 years	48	(17.0)	17	(18.1)	31	(16.5)	Ref. (0.433–2.308)	0.4613
51–60 years	97	(34.4)	37	(39.4)	60	(31.9)	1.125 (0.548–2.309)	*>0.9999 ^B^*
61–70 years	98	(34.8)	27	(28.7)	71	(37.8)	0.693 (0.331–1.452)	
>70 years	39	(13.8)	13	(13.8)	26	(13.8)	0.912 (0.374–2.222)	
Smoking								
Never smoker	57	(20.2)	16	(17.0)	41	(21.8)	Ref. (0.442–2.264)	**<0.0001**
Former Smoker	100	(35.5)	10	(10.6)	90	(47.9)	0.285 (0.119–0.681)	** *<0.0001 ^B^* **
Current Smoker	125	(44.3)	68	(72.3)	57	(30.3)	3.057 (1.554–6.013)	
Pack years (Quartiles)								
0 PY	57	(20.2)	16	(17.0)	41	(21.8)	Ref. (0.442–2.264)	0.1154
>0–20 PY	82	(29.1)	22	(23.4)	60	(31.9)	0.940 (0.441–2.002)	*>0.9999 ^B^*
21–38 PY	70	(24.8)	24	(25.5)	46	(24.5)	1.337 (0.625–2.858)	
>38 PY	73	(25.9)	32	(34.0)	41	(21.8)	2.000 (0.954–4.192)	
Pack Years ^b^								
Mean (95% CI)			29.88	(24.73–35.03)	24.40	(20.34–28.47)		0.1035 ^b^
Fagerström Test for Nicotine Dependence								
Low (Score 0–2)	203	(73.3)	51	(54.3)	152	(83.1)	Ref. (0.639–1.566)	**<0.0001**
Low to Moderate (Score 3–4)	28	(10.1)	17	(18.1)	11	(6.0)	4.606 (2.024–10.48)	** *0.0012 ^B^* **
Moderate (Score 5–7)	38	(13.7)	22	(23.4)	16	(8.7)	4.098 (1.999–8.401)	
High (8–9)	8	(2.9)	4	(4.3)	4	(2.2)	2.980 (0.719–12.35)	
Fagerström Test ^b^								
Mean (95% CI)			2.59	(2.08–3.09)	1.03	(0.71–1.34)		**<0.0001 ^b^**
Alcohol Use								
Never drinker	39	(13.8)	9	(9.6)	30	(16.0)	Ref. (0.349–2.868)	**<0.0001**
Former Drinker	12	(4.3)	12	(12.8)	0	(0)	80.26 (4.333–1486.8) ^c^	** *0.0004 ^B^* **
Current Drinker	231	(81.9)	73	(77.7)	158	(84.0)	1.540 (0.696–3.410)	
Daily Amount of Alcohol								
0 or <1 g/d	39	(13.8)	9	(9.6)	30	(16)	Ref. (0.349–2.868)	**<0.0001**
1–30 g/d	84	(29.8)	31	(33.0)	53	(28.2)	1.950 (0.819–4.639)	** *0.0035 ^B^* **
31–60 g/d	43	(15.2)	27	(28.7)	16	(8.5)	5.625 (2.136–14.81)	
>60 g/d	116	(41.1)	27	(28.7)	89	(47.3)	1.011 (0.428–2.391)	
AUDIT Test for Alcohol Dependence								
High Dependence (≥8)	60	(21.3)	19	(20.2)	41	(21.8)	0.908 (0.493–1.673)	0.7576
Low Dependence (<8)	222	(78.7)	75	(79.8)	147	(78.2)		*>0.9999 ^B^*
AUDIT Test ^b^								
Mean (95% CI)			5.07	(4.03–6.12)	5.44	(4.76–6.12)		0.5640 ^b^
Lifetime no. of vaginal-sex partners								
0–5	170	(60.3)	60	(63.8)	110	(58.5)	Ref. (0.641–1.560)	0.6187
6–25	95	(33.7)	28	(29.8)	67	(35.6)	0.766 (0.446–1.317)	*>0.9999 ^B^*
>25	17	(6.0)	6	(6.4)	11	(5.9)	1.000 (0.352–2.839)	
Lifetime no. of oral-sex partners								
0	143	(50.7)	59	(62.8)	84	(44.7)	Ref. (0.624–1.601)	**0.0151**
1–5	103	(36.5)	27	(28.7)	76	(40.4)	0.506 (0.292–0.878)	*>0.9999 ^B^*
≥6	36	(12.8)	8	(8.5)	28	(14.9)	0.407 (0.173–0.955)	
0	143	(50.7)	59	(62.8)	84	(44.7)	Ref. (0.624–1.601)	**0.0042**
any oral sex (>0)	139	(49.3)	35	(37.2)	104	(55.3)	0.479 (0.288–0.796)	*0.7162 ^B^*
Casual-sex partner ^d^								
Yes	109	(38.7)	34	(36.2)	75	(39.9)	0.854 (0.512–1.425)	0.5450
No	173	(61.3)	60	(63.8)	113	(60.1)		*>0.9999 ^B^*
Age at first intercourse								
<18 Years	127	(45.0)	55	(58.5)	72	(38.3)	2.272 (1.372–3.764)	**0.0013**
≥18 Years	155	(55.0)	39	(41.5)	116	(61.7)		*0.2222 ^B^*
Condom use								
Never or rarely	244	(86.5)	79	(84.0)	165	(87.8)	0.734 (0.363–1.484)	0.3880
Usually or always	38	(13.5)	15	(16.0)	23	(12.2)		*>0.9999 ^B^*
Sexually transmitted disease								
Yes	24	(8.5)	4	(4.3)	20	(10.6)	0.373 (0.124–1.126)	0.0702
No	258	(91.5)	90	(95.7)	168	(89.4)		*>0.9999 ^B^*
Oral warts or papilloma								
Yes	2	(0.7)	1	(1.1)	1	(0.5)	2.000 (0.124–32.334)	0.6186
No	279	(99.3)	93	(98.9)	186	(99.5)		*>0.9999 ^B^*
Genital warts								
Yes	8	(2.8)	4	(4.3)	4	(2.1)	2.033 (0.497–8.318)	0.3142
No	273	(97.2)	90	(95.7)	183	(97.9)		*>0.9999 ^B^*
Sexual partner with genital warts								
Yes	6	(2.2)	0	(0)	6	(3.2)	0.148 (0.008–2.650) ^c^	0.0791
No	272	(97.8)	93	(100)	179	(96.8)		*>0.9999 ^B^*
First-degree relative with SCC ^e^								
Yes	8	(3.0)	4	(4.6)	4	(2.2)	2.096 (0.512–8.590)	0.2937
No	257	(97.0)	83	(95.4)	174	(97.8)		*>0.9999 ^B^*
First-degree relative with cancer at any site								
Yes	99	(35.5)	34	(36.6)	65	(34.9)	1.073 (0.639–1.802)	0.7907
No	180	(64.5)	59	(63.4)	121	(65.1)		*>0.9999 ^B^*

^a^ The *p*-values are from the *Pearson’s Chi-square* (*χ*^2^) test. *^B^ Bonferroni*-corrected *p*-values for multiple comparisons *italic.*
^b^ The *p*-value is from the heteroscedastic *t* test. ^c^ Odds ratio adjusted according to *Cox* and *Moses* by adding 0.5 to each cell to reduce bias and prevent division by zero caused by empty cells [12,13]. ^d^ A casual-sex partner was defined as a partner in a “one-night stand” or a partner who was a stranger. ^e^ Squamous cell carcinoma. ^#^ OPSCC patients who could not be matched with two appropriate controls having a propensity score within the individuals PS ± 0.1 were excluded.

**Table 3 cancers-15-03356-t003:** Comparison between self-reported numbers of lifetime vaginal-sex and oral-sex partners within controls in the study of D’Souza et al. and our PS-matched sample.

	Total		D’Souza et al. [1]		LIFE A1			
Controls	*n*	(%)	*n*	(%)	*n*	(%)	OR (95% CI)	*p*-Value ^a^
Lifetime no. of oral-sex partners								
0	122	(31.4)	38	(19.0)	84	(44.7)	3.443 (2.184–5.43)	**<0.0001**
≥1	266	(68.6)	162	(81.0)	104	(55.3)		
0–5 partners	308	(79.4)	148	(74.0)	160	(85.1)	2.008 (1.205–3.347)	**0.0068**
≥6 partners	80	(20.6)	52	(26.0)	28	(14.9)		
Lifetime no. of vaginal-sex partners								
0–5 partners	218	(56.2)	108	(54.0)	110	(58.5)	1.201 (0.804–1.796)	0.3708
≥6 partners	170	(43.8)	92	(46.0)	78	(41.5)		
<26 partners	348	(89.7)	171	(85.5)	177	(94.1)	2.729 (1.321–5.635)	**0.0051**
≥26 partners	40	(10.3)	29	(14.5)	11	(5.9)		

^a^ The *p*-values are from *Pearson’s Chi-square* (*χ*^2^) tests.

**Table 4 cancers-15-03356-t004:** Comparison between self-reported numbers of lifetime vaginal-sex and oral-sex partners within patients with OPSCC in the study of D’Souza et al. and our PS-matched sample.

Cases	Total		D’Souza et al. [1]		LIFE B7 OPSCC			
	*n*	(%)	*n*	(%)	*n*	(%)	OR (95% CI)	*p*-Value ^a^
Lifetime no. of oral-sex partners								
0	71	(36.6)	12	(12.0)	59	(62.8)	12.362 (5.934–25.753)	**<0.0001**
≥1	123	(63.4)	88	(88.0)	35	(37.2)		
0–5 partners	144	(74.2)	58	(58.0)	86	(91.5)	7.784 (3.407–17.784)	**<0.0001**
≥6 partners	50	(25.8)	42	(42.0)	8	(8.5)		
Lifetime no. of vaginal-sex partners								
0–5 partners	91	(46.9)	31	(31.0)	60	(63.8)	3.928 (2.162–7.137)	**<0.0001**
≥6 partners	103	(53.1)	69	(69.0)	34	(36.2)		
<26 partners	160	(82.5)	72	(72.0)	88	(93.6)	5.704 (2.239–14.530)	**<0.0001**
≥26 partners	34	(17.5)	28	(28.0)	6	(6.4)		

^a^ The *p*-values are from *Pearson’s Chi-square* (*χ*^2^) tests.

## Data Availability

The complete datasets presented in this article are not readily available because of patient confidentiality and participant privacy terms. Requests to access the datasets should be directed to G.W., gunnar.wichmann@medizin.uni-leipzig.de.

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
