# Peer review of "Is High-Risk Sexual Behavior a Risk Factor for Oropharyngeal Cancer?"

_cancers, 2023, doi:10.3390/cancers15133356_

Round 1
Reviewer 1 Report
This manuscript “Is high-risk sexual behaviour a risk factor for oropharynx cancer?” by Gunnar Wichmann and colleagues reports an investigation aimed to replicate early evidence by D’Souza et al. in a different population. The rationale is scientifically relevant and the methodological approach, which includes an accurate and rigorous selection of population-based healthy controls, represents a key factor in increasing the quality of the investigation. However, several major concerns lowered the quality of the research and the appeal of the manuscript.
The main point of this investigation is the selection of a matched control sample using the propensity score (PS). Surprisingly, almost no information about PS was provided. It is strongly required a detailed paragraph in the Method section about this topic. How the score was calculated, which covariates were accounted for, how data for each covariate were entered, how in case of missing data, how matching was done, and so on.
It is not clear how PS was used for the selection of controls described in Table 1. Moreover, it is shocking to see how this process failed to eliminate or mitigate the effect of any of the confounding biases. Indeed, each single covariate distribution strongly differed between HNSCC cases and controls, with p values < .0001 in each case.
A more stringent matching procedure mitigated the problem but did not solve it completely (Table 2). Why did the authors not apply the most effective method for both instances, i.e., HNSCC and OFSCC? Considering that the average reader could be unfamiliar with the effectiveness of PS-matched control selection in reducing the effect of covariates, the authors need to discuss this subject.
There are several inconsistencies in data reported in the Results section. It follows some examples.
Lines 190-191 “According to Table 2, the PS-matched OPSCC subgroup and their PS-matched controls no longer deviated significantly as they demonstrated the comparable distribution of sex”. This is incorrect because the test for equality of sex distribution in Table 2 produced a p-value = 0.0035.
Lines 198-200 “Within the PS-matched analysis, we found OPSCC patients being more likely to self report their first sexual intercourse before age 18, but also using a condom usually or always. Sexually transmitted diseases (STD) were more frequent among controls.” This is incorrect because the test for condom use provided equal distribution (p-value = 0.38), and the test for sexually transmitted diseases was also not significant.
Lines 201-205 There are other incorrect data reporting, including the false statement of no association with oral sex partners. This is awful because it contrasts with the main conclusion of this investigation.
The first paragraph of the Result section is very confusing; the authors have to rewrite it.
As regard Figure 2: the legend is almost incomprehensible; the percentage calculation reported in panels b, c, e, and f is meaningless. If unaffected cases with no HR-SB are relevant, they need to be included in Venn's four-set diagram.
Logistic regression is a complex and powerful analytic method that can manage one or more independent variables and covariate adjustment. A single sentence like that reported in lines 208-210 is worthless.
The Discussion section is too long.
The in deep comparison of the authors’ data with D’Souza’s data as made in Tables 3 and 4 is inopportune and wrong. In particular, the comparison of the unaffected controls reported in Table 3 is meaningless. Indeed, neither one of the two groups is representative of the general population: the D’Souza group because of the reasons well detailed in the manuscript and the Life A1 because it is selected to match the PS of OFSCC patients.
Author Response
Dear reviewer, please find the responses to your comments in the PDF file submitted.
Many thanks for your very helpful suggestions!

Reviewer 2 Report
I totally agree with the authors that more epidemiologic data on the subject are needed, and that the design of their study helps to avoid most bias. This study can open further questions for HR-SB in these cancer patients.
Few points to improve:
1/ The authors should better highlight the principal limitation: using interviews. This definitely explains the "absence" of anal sex in their cohort. That could also mean biased responses to all other questions. The fact that "live" interviews were used should be better highlighted.
2/ The controls had no HNSCCs or other cancers? To define in MM.
3/ Given the age of the population studied, HPV vaccination was probably not perfomed, but the absence of this info should be noted. Similarly, epidemiological studies in the status of OPSCC in the after HPV vaccine era (especially in some countries like Portugal where the vaccination of both girls and boys is obligatory since several years now) should be provided.
4/ Could the authors define the localizations of oropharyngeal cancers?
5/ It is not at all clear the differences between the initial HNSCC cohort and the control given the PS match. The authors should explain this.
6/ This is also true for the female male difference between the final oropharyngeal cohort and the control. Can the authors provide data for men/women only? Having 20% female patients is an important part for such a small cohort and definitely associated with different oral sex habits between women and men. The authors should comment on it by using literature data.
6/ Apart the study of D'Souza, other studies on the subject? More recent also? 2007 seems far away, so much progress has been seen in ten years..
7/ I think something is wrong with the ref in the phrase "Maura Gillison replied [18] to a comment [18] to their report"
8/ Given that much of their intro/discussion is devoted to case-control studies quality, can the authors discuss similar studies in the other HPV driven cancers especially cervical, anal?
Author Response

(The authors gave the same response as above.)

Reviewer 3 Report
comments:
1. On Table 1, is it supposed that young people have more high-risk behavior? so why age group is from 18-50? how many from age 18-30? for example.
2. On Table 1, more details on BMI, lipid panel, and special nuts use? Betel nuts?
3. Any significant difference between sex groups on p16 staining (positivity)?
4. How many on lifetime number of vaginal sex partners 0-2, and 2-5? on page 9.
5. definition of casual sex-partner?
Author Response

(The authors gave the same response as above.)

Round 2
Reviewer 1 Report
This reviewer read the Authors response with attention. Several pieces of information need to be included in the manuscript.
The control selection by propensity scores the good of the research plan. However, because of a series of reasons mentioned in the response letter, the authors failed to identify matched controls. The reader need to be informed of this problem because is a serious limitation of this investigation.
The authors forget to revise the first paragraph of the results section as stated.
Figure 2 needs a Venn's four-set diagram (https://bioinfogp.cnb.csic.es/tools/venny/) and a readable legend.
Author Response
Dear reviewer #1,
many thanks again for the very helpful suggestions and comments helping us to improve the paper and in particular to the very welcome discussion of the problems in obtaining sufficient controls for sample of patients with epidemiologic characteristics and lifestyle-associated risk factors not easily to find in a population-based sample! Please find the response to your suggestions and comments in the uploaded PDF.
Best regards
Gunnar Wichmann

Reviewer 3 Report
No more comments
Author Response
Dear Reviewer 3,
many thanks for your time and your helpful comments for revising the manuscript.
Sincerely yours,
Gunnar Wichmann